# Myeloperoxidase Inhibition Ameliorates Plaque Psoriasis in Mice

**DOI:** 10.3390/antiox10091338

**Published:** 2021-08-25

**Authors:** Savannah D. Neu, Anna Strzepa, Dustin Martin, Mary G. Sorci-Thomas, Kirkwood A. Pritchard, Bonnie N. Dittel

**Affiliations:** 1Versiti Blood Research Institute, Milwaukee, WI 53201, USA; sneu@mcw.edu (S.D.N.); strzepa@gmail.com (A.S.); 2Department of Microbiology & Immunology, Medical College of Wisconsin, Milwaukee, WI 53226, USA; 3Department of Medical Biology, Jagiellonian University Medical College, 31-034 Krakow, Poland; 4Department of Surgery, Division of Pediatric Surgery, Medical College of Wisconsin, Milwaukee, WI 53226, USA; dpmartin0012@gmail.com (D.M.); kpritch@mcw.edu (K.A.P.J.); 5Department of Medicine, Division of Endocrinology & Molecular Medicine, Medical College of Wisconsin, Milwaukee, WI 53226, USA; msthomas@mcw.edu

**Keywords:** plaque psoriasis, myeloperoxidase, neutrophils, Imiquimod, Aldara, psoriasis area and severity index (PASI), N-acetyl lysyltyrosylcysteine amide (KYC)

## Abstract

Plaque psoriasis is a common inflammatory condition of the skin characterized by red, flaking lesions. Current therapies for plaque psoriasis target many facets of the autoimmune response, but there is an incomplete understanding of how oxidative damage produced by enzymes such as myeloperoxidase contributes to skin pathology. In this study, we used the Aldara (Imiquimod) cream model of plaque psoriasis in mice to assess myeloperoxidase inhibition for treating psoriatic skin lesions. To assess skin inflammation severity, an innovative mouse psoriasis scoring system was developed. We found that myeloperoxidase inhibition ameliorated psoriasis severity when administered either systemically or topically. The findings of this study support the role of oxidative damage in plaque psoriasis pathology and present potential new therapeutic avenues for further exploration.

## 1. Introduction

Plaque psoriasis is an inflammatory condition that causes redness and flaking of the skin. Acute plaque psoriasis is characterized by short-term skin lesion formation; known as a relapse; triggered by allergens, medication, or infection which resolves over time [1]. The etiology of recurrent plaque psoriasis is incompletely understood but is thought to involve dysregulation of immune cells [2]. Around 1–3% of people experience plaque psoriasis (acute or chronic) during their life and the resulting skin lesions are itchy, painful, and can cause emotional distress [3,4]. Therapies, both topical and systemic, are used to hasten recovery, manage symptoms and discomfort, and reduce the frequency of relapses, but there is no cure for plaque psoriasis [5,6].

When psoriatic skin lesions form, fluid and immune cells are recruited to the dermal layers of the skin which leads to initial swelling and reddening [7]. The immune cells secrete factors that promote hypertrophy of the epithelium (thickening) and recruit further effector immune cells [8,9,10]. During the most severe stages of plaque psoriasis, the outermost layers of the hypertrophied epithelium begin to dry out and flake off (scaling), leaving the underlying layers exposed and sensitive to temperature and touch [11,12].

Many studies have highlighted the pro-inflammatory contribution of T cells in plaque psoriasis, demonstrating the potentially autoimmune component of this condition [13,14,15]. However, T cell involvement in lesion formation is often preceded by the recruitment of innate immune cells such as neutrophils [16]. Neutrophils are highly abundant in the blood and are rapidly recruited to sites of developing inflammation and can destroy pathogens via multiple cytotoxic mechanisms [17,18]. Through a process known as degranulation, neutrophils secrete large volumes of degradative enzymes and inflammatory signals; and by NETosis extrude DNA to trap infectious agents and clear debris [19,20]. The presence of neutrophils in plaque psoriasis lesions has been documented and depleting neutrophils improved skin inflammation recovery in animal models [21,22,23]. However, the mechanisms whereby neutrophils contribute to plaque psoriasis pathology are not well understood.

Myeloperoxidase (MPO), a heme peroxidase, is the most abundant enzyme found in neutrophilic granules and is cytolytic [24]. In the inflammatory microenvironment where radical oxygen species are plentiful, MPO “bleaches” cells and proteins through a process involving the breakdown of hydrogen peroxide and the generation of hypochlorous acid [25,26,27]. In addition to microbicidal activity, MPO can cause extensive tissue damage under sterile conditions as seen in other animal models of autoimmune inflammation [28]. Along with neutrophils, MPO is secreted by macrophages which can surveil the skin [29,30].

Although studies have increasingly suggested that oxidative damage plays a role in plaque psoriasis, it is not known to what extent MPO contributes to overall oxidative stress in this disease [31,32]. There are currently no FDA-approved MPO inhibitors available. To overcome that hurdle, we designed N-acetyl lysyltyrosylcysteine amide, a novel tripeptide (Lys-Tyr-Cys (KYC)) inhibitor of MPO toxic oxidant production [33]. KYC is non-toxic in mice and in previous studies, we showed that it attenuated the severity of a variety of inflammatory animal models of disease including multiple sclerosis, contact hypersensitivity, stroke and sickle cell disease [34,35,36,37,38].

Here, we asked whether MPO plays a pathogenic role in plaque psoriasis using the Imiquimod (IMQ) mouse model that induces an innate immune response in the skin via its activity as a toll-like receptor (TLR) 7 agonist [39]. To assess the severity of plaque psoriasis, we generated a modified human psoriasis area and severity index (PASI) for mice assessing the severity of erythema, induration and desquamation in IMQ-induced skin lesions. We found that both MPO deficiency and inhibition resulted in attenuated plaque psoriasis severity. Of importance for therapeutic use, topical application of the MPO inhibitor KYC reduced the PASI score as soon one day later. These findings suggest that MPO-targeted therapeutics may be efficacious for the treatment of plaque psoriasis in humans.

## 2. Materials and Methods

### 2.1. Mice

C57BL/6 mice were purchased from The Jackson Laboratory (Bar Harbor, ME, USA). C57BL/6-*Mpo^tm1Lus^* (*Mpo^−/−^*), originally purchased from The Jackson Laboratory, were bred and maintained in-house. All animals were housed in the Translational Biomedical Research Center of the Medical College of Wisconsin (MCW). Purchased mice were allowed to acclimate to the animal facility for at least one week and included in experiments starting at eight weeks of age. All mouse handling and care procedures were approved by the MCW IACUC.

### 2.2. Shaving

In preparation for plaque psoriasis induction on the back, a rectangular area of 9 cm^2^ was shaved from between the shoulder blades down to within one centimeter of the base of the tail. Throughout shaving, animals were anesthetized using an isoflurane respirator and back fur was wetted with warm water. The fur was first trimmed with an electric clipper marketed for animal grooming and then the remaining stubble was removed with a razor blade. Razor blades were replaced with every other mouse. Hair around the shaved region was trimmed to reduce cowlick formation. Mice were rested without handling for one or two days before disease induction. Mice with incomplete fur removal were excluded from experiments.

### 2.3. Plaque Psoriasis Induction

To induce plaque psoriasis, Aldara IMQ cream, 5% (3M Pharmaceuticals, St. Paul, MN, USA) was applied to the skin. IMQ or a control moisturizer cream was applied to the front and back sides of one ear or to shaved back skin. While mice were anesthetized, creams were applied daily in 50 mg doses using a metal spreader. After cream application, mice were monitored in a bedding-less cage for at least ten minutes to interrupt licking behaviors and allow the cream to dry. Creams were applied for five consecutive days until plaque psoriasis peaked and then the cream application was terminated to allow symptom resolution [40].

### 2.4. KYC Treatment

To treat plaque psoriasis, mice were given doses of KYC either systemically or through topical application. Systemic doses of KYC were chosen based on previous studies [35]. For systemic administration, KYC, synthesized by the Versiti Blood Research Institute Protein Core Lab, was solubilized in PBS to form stock concentrations of 15 mg/mL, aliquoted, and then stored at −20 °C for up to two weeks. A day prior to use, stock KYC was subsequently diluted in PBS to achieve treatment doses and stored at 4 °C for up to one week. Intraperitoneal (IP) injections of KYC in 100 μL were given daily from days 0–6 just prior to IMQ cream application. Mice received KYC at doses of 0.3 or 3 mg/kg based on average bodyweight of the mice [41]. Control, untreated mice received PBS injections alone. For topical administration, ultra-pure KYC (HPLC-purified) (MCW Protein and Nucleic Acid Core Lab) was used to minimize side effects of trifluoracetic anhydride used in KYC synthesis on the irritated skin. Ultra-pure KYC was dissolved in water to form stock concentrations of 5 mg/mL, aliquoted, and then stored at −20 °C long-term. A day prior to use, stock ultra-pure KYC was diluted into 100 μL of water and then mixed into 500 mg of Sorbolene cream; a water-based, skin moisturizer [42]. Ultra-pure KYC creams were hand-mixed for five minutes with a metal spatula and then stored at 4 °C for up to five days. Complete homogenization of dissolved solute into Sorbolene cream was verified using a water-based dye in test mixings. To treat plaque psoriasis, 50 mg of KYC cream containing either 30 or 50 μg KYC was applied daily to the skin using a metal spatula. KYC cream was applied after signs of plaque psoriasis began through symptom resolution. KYC cream was administered eight hours after IMQ cream to minimize treatment mixing. No-disease and IMQ control mice received Sorbolene cream containing water alone.

### 2.5. Ear Inflammation Quantification

In experiments where plaque psoriasis was induced on the ears, ear swelling was measured using calipers while mice were anesthetized. Calipers were aligned to the center of the ear immediately distal to the scapha. Both the left and right ears were measured prior to psoriasis induction to serve as the baseline thickness. Aldara cream was applied for four consecutive days to both sides of the right ear. On day five after induction start, ear thickness was measured again and baseline thickness was subtracted. Three independent measurements were recorded per ear per time point and individual data points are shown as the average swelling of each ear.

### 2.6. Plaque Psoriasis Severity Scoring

In experiments where plaque psoriasis was induced on the back, skin inflammation was quantified using a novel scoring system developed for this project that assessed erythema (redness), induration (thickness) and desquamation (scaling). To score plaque psoriasis in mice, skin parameters were scored daily on a scale of 0 (no incidence of disease) to 4 (severe). Percent body surface was not included in scoring because the psoriatic lesions were sequestered to the same 9 cm^2^ area of the back. Appearance of the back skin was compared to example images compiled from preliminary plaque psoriasis testing. The three parameter scores were then summed to generate the daily cumulative score for each mouse. Pictures of the back skin were taken for record-keeping using a 16-megapixel camera. Pictures of the skin were cropped to center the shaved skin region in the frame of reference, but no other alterations were made.

### 2.7. Histology

At the peak of plaque psoriasis severity (day 3), some mice were euthanized to collect samples of the back skin. One cm long skin segments were excised from the shaved back region while the skin was relaxed. Skin segments were embedded in TissueTek O.C.T. compound oriented vertically with the longest edge facing down and then frozen on dry ice as previously described [43,44]. A total of 10 μm sections were H&E stained. Skin sections were assessed for the overall level of skin inflammation via thickening of the epithelium and infiltration of immune cells into the dermis.

### 2.8. Statistical Analysis

Plaque psoriasis disease course was assessed using the Friedman test (non-parametric, repeated measures) with accompanying group comparison. The area under the curve (AUC) was calculated for each mouse following the trapezoid rule. Peak severity was chosen as the experimental day when most animals had their highest disease incidence (day 3 or 4). AUC and peak severity values were compared between experimental groups using the Kruskal–Wallis test with multiple comparisons. Statistical significance was considered as having a confidence interval of 95% (*p* ≤ 0.05). Data are shown graphically as individual points that represent one animal or ear and as bar charts constituting the group mean. Error bars signify the standard error of the mean (SEM). Statistical testing and graphing were performed using GraphPad Prism v9.0.

## 3. Results

### 3.1. MPO Deficiency Attenuated Plaque Psoriasis

To address whether MPO plays a role in plaque psoriasis, psoriasis was induced on the ears of wildtype and MPO-deficient (*Mpo^−/−^*) mice using Aldara (IMQ) cream. Aldara cream is a topical skin cancer and keratosis treatment that contains IMQ, a toll-like receptor 7 agonist [45,46]. When applied to the skin, IMQ causes acute, inflammatory skin lesioning that resembles plaque psoriasis pathology [40,47]. MPO deficiency did not alter the steady-state ear thickness of non-induced ears as compared to WT mice (Figure 1). Both wildtype and *Mpo^−/−^* mice experienced site-specific ear inflammation as evidenced by negligible swelling of the non-induced left ear control (Figure 1). In contrast, *Mpo^−/−^* mice exhibited significantly reduced swelling of the right ear upon IMQ application in comparison to WT controls (Figure 1).

### 3.2. Development of a Mouse PASI Scoring System

Human plaque psoriasis is clinically assessed using the psoriasis area and severity index (PASI) scoring system by visually scoring three parameters of epidermal appearance: erythema (redness), induration (thickness), and desquamation (scaling or flaking); as well as overall (%) body surface area affected [48,49,50]. To simulate the human PASI system in our mouse model, plaque psoriasis was induced on the backs of wild-type mice and tracked over the course of the disease (days 0–6). Though both male and female mice were tested in initial experiments, only female mice were used in further testing due to their less aggressive behavior. Prior to experimentation, shredded paper for nesting was removed from mouse cages to protect exposed back skin from rubbing irritation. Mice were divided into groups of 3–5 animals/cage. Each cage comprised a single experimental group given the same cream applications to avoid dose mixing. Mice that developed severe skin lesions (oozing wounds, broken skin) or formed hunched backs were removed from experimentation for humane reasons. This pathology was uncommon (<1 animal per experiment) and not associated with the normal development of induced plaque psoriasis. From these data, a mouse PASI scoring scale was developed to aid in reproducibly assessing mouse plaque psoriasis inflammation on the back skin (Table 1, Figure 2, Appendix A). Representative photographic images for each parameter score were compiled from three replicate experiments (Figure 2). Artistic representations of each stage of psoriasis were also generated for assistance with assigning a score (Appendix A). Images used for induration and desquamation were desaturated to minimize visual bias from erythema. Individual descriptions for each parameter and severity are included in Table 1. For each skin parameter, a severity score of four was determined from the most severe cases experienced in these preliminary tests. Every animal given IMQ cream developed plaque psoriasis skin lesioning, but no animal ever achieved a score of four in all parameters at a single time point.

### 3.3. Systemic MPO Inhibition Attenuated Plaque Psoriasis

To address whether MPO inhibition could attenuate plaque psoriasis, inflammation was induced on the backs of mice with IMQ and treated with KYC systemically. Mice were administered daily IMQ cream from days 0–5 and daily IP injections of KYC from days 0–6 (Figure 3A). Signs of plaque psoriasis were seen as early as day 1, peaked on day 4, and began to resolve by day 6 (Figure 3B). Mice that received KYC treatment experienced attenuated daily mouse PASI scores from days 2–5 in a dose-dependent manner (Figure 3B). To assess statistical differences in the cumulative disease incidence between treatment groups, AUC was calculated. A small number of mice experienced skin inflammation from shaving as seen in the cream controls (Figure 3C). Although psoriasis was attenuated daily by KYC treatment, the cumulative disease index (AUC) was only significantly reduced in the 3 mg/kg dose group (Figure 3C). However, treatment with either dose of systemic KYC reduced peak plaque psoriasis severity (day 3) (Figure 3D) and visually improved skin appearance (reduced redness, thickening, and flaking) (Figure 3E).

### 3.4. Topical Administration of KYC Attenuated Plaque Psoriasis

To address whether topical KYC would be an efficacious therapeutic strategy for plaque psoriasis, IMQ was applied to the shaved back and then affected skin was directly treated with cream containing KYC. Mice were administered daily IMQ cream from days 0–4 (Figure 4A). KYC cream was administered after signs of psoriasis were first observed, which occurred as early as 12 h after induction through day 7 (Figure 4B). Mice that received cream containing 30 μg KYC experienced reduced daily mouse PASI scores from days 3–5 (Figure 4B). Mice that received cream containing 50 μg KYC experienced reduced daily mouse PASI scores from days 2–6 (Figure 4B). To assess differences in the cumulative disease incidence, AUC was calculated and both topical KYC treatments significantly attenuated plaque psoriasis over the course of the disease (Figure 4C). At peak of disease (day 3), treatment with topical KYC cream significantly reduced mouse PASI scores in a dose-dependent manner (Figure 4D). Visual assessment of plaque psoriasis at peak severity (day 3) showed attenuation of all three skin parameters (redness, thickness, flaking) during KYC cream treatment (Figure 4E).

To better understand how topical KYC treatment attenuated plaque psoriasis, the three parameters of skin inflammation were assessed individually. Composite daily mouse PASI scores shown in Figure 4B were divided into individual parameter scores: erythema (Figure 5A–C), induration (Figure 5D–F) and desquamation (Figure 5G–I). KYC treatment ameliorated daily skin reddening on days 2–4 (Figure 5A). However, the overall erythema incidence across the disease course was not significantly reduced in either treatment group (Figure 5B). Treatment with 50 μg KYC attenuated erythema at peak (day 3) psoriasis, but treatment with 30 μg KYC only trended towards significance (*p* = 0.07) (Figure 5C). Treatment with 50 μg KYC attenuated daily skin induration on days 2–3 (Figure 5D). However, neither KYC dose reduced the overall induration incidence (Figure 5E) or peak induration (Figure 5F). Finally, both KYC treatments significantly reduced the daily level of desquamation from days 2–6 (Figure 5G), as well as the overall desquamation incidence (Figure 5H) and peak (day 3) desquamation severity (Figure 5I).

To visualize the impact of KYC treatment on plaque psoriasis pathology, skin histology was performed at the peak of disease (day 3) on control and topical KYC cream-treated mice. Mice received either control creams only, IMQ alone, or IMQ with 50 μg KYC cream for three days. Comparing H&E-stained sections of healthy (Figure 6A) and lesioned (Figure 6B) skin showed that application of IMQ cream led to epithelial hypertrophy and increased infiltration of immune cells into the dermis. In the epidermal layer, immune infiltrates in IMQ-treated skin appeared to contain mostly neutrophils (Figure 6B) which could not be found in the cream control (Figure 6A). Topical treatment with 50 μg KYC treatment reduced epithelial hypertrophy and incidence of immune infiltration from IMQ (Figure 6C). Additionally, neutrophil-appearing cells were rare (Figure 6C).

## 4. Discussion and Conclusions

Here, we found that MPO deficiency attenuated plaque psoriasis pathology in the IMQ mouse model (Figure 1). However, because the *Mpo^−/−^* mouse is a global genetic knockout, it limited our ability to separate contributions of MPO oxidative damage from whole-body, long-term effects due to MPO-deficiency. Therefore, to overcome these limitations, we transitioned the experimental strategy to using targeted MPO inhibition via KYC in wild-type mice. In studies previously conducted, we found that systemic administration of KYC attenuated both T cell-mediated and other autoimmune, inflammatory conditions where neutrophil and MPO activity was believed to occur [35,36,37,38]. In the present study, we administered KYC systemically through IP injection and found that overall plaque psoriasis severity was attenuated in a dose-dependent manner (Figure 4). These results support targeting MPO oxidative stress pathways for the treatment of plaque psoriasis and serve as proof-of-concept for using MPO inhibition as a therapeutic strategy.

While applying IMQ to the ears of a mouse was a straightforward and highly quantifiable psoriasis induction strategy, it did not yield sufficient information on the extent or quality of psoriatic skin lesions for evaluating innovative treatments (Figure 1). Therefore, to better assess the therapeutic potential of MPO inhibition via KYC, we induced psoriasis on the shaved backs of mice. Other skin-focused studies often use hairless nude mice, but we chose to use wild-type black mice to eliminate potential confounding genetic effects from the nude mouse background [51,52]. Using hairy mice presented a need for an additional shaving step, but shaving did not seem to impact psoriasis incidence or development (Figure 3). For this experimental strategy, we also generated a novel scoring method to track skin inflammation on the mouse back (Figure 2). The mouse PASI scoring system, based on the human psoriasis scoring system, could reproducibly differentiate the severity of plaque psoriasis lesions based on three characteristics of skin appearance: redness/erythema, thickening/induration, and flaking/desquamation (Figure 2, Table 1). In our IMQ psoriasis testing, most mice experienced moderate to severe erythema and induration with fewer animals experiencing severe desquamation (Figure 5). Skin reddening was usually the first symptom to contribute to the overall disease score, followed by thickening and then flaking (Figure 5). Redness and thickening subsided around day 6, but skin flaking sometimes persisted beyond Day 7 as lesions continued to heal (Figure 5). The mouse PASI scoring system was developed with the IMQ mouse model in mind but could be applied to other models of skin inflammation where redness and flaking occur [53].

Although systemic administration of KYC through IP injection attenuated IMQ psoriasis (Figure 4), this therapeutic route is not used to treat human illness. Systemic dosing of KYC to continuously target MPO oxidative stress could be achieved through devices such as an implanted Alzet pump; however, these devices are mildly invasive and require routine maintenance [54,55,56]. While our previous studies supported that systemic KYC is not toxic, whole-body treatment may have off-target effects when MPO is inhibited beyond the skin [57]. As mentioned previously, MPO is microbicidal, so systemic MPO inhibition may negatively impact immune responses to clear infections [58,59]. Therefore, to reduce these off-target effects, we tested KYC administration through a topical cream treatment that could be site-directed to the affected skin on the back (Figure 4). Currently used topical treatments for plaque psoriasis include over-the-counter steroids and chemical agents that protect the epidermis, but other treatment targets are severely lacking [60,61]. To better mirror human plaque psoriasis topical therapies, KYC creams were applied only after signs of psoriasis began (Figure 4). Overall, topical KYC treatment attenuated IMQ psoriasis lesioning when given once a day (Figure 4). Although the exact mechanism whereby MPO inhibition ameliorated plaque psoriasis is unknown, our data showed that KYC treatment attenuated flaking severity more than any other parameter of skin inflammation (Figure 5). In accordance with this, histology of skin lesions suggested that epithelial hypertrophy was reduced when IMQ psoriasis was treated with the most efficacious daily topical KYC cream treatment containing 50 μg KYC (Figure 6). Because neutrophils were rare in KYC-treated skin lesions (Figure 6C) compared with untreated IMQ lesions (Figure 6B), neutrophil activities such as MPO secretion may contribute to plaque psoriasis by promoting skin epithelium overgrowth which leads to eventual desquamation. Although our studies emphasize neutrophils, macrophages could also contribute to MPO activity. Furthermore, topical KYC creams could be administered at varying doses and multiple times a day, providing a means to tailor the therapy for each patient’s needs.

Overall, our data provided a valuable proof-of-concept for targeting oxidative stress pathways to treat plaque psoriasis. MPO deficiency and inhibition through KYC ameliorated skin inflammation severity in the IMQ mouse model, evidencing the contribution of MPO and oxidative damage during acute plaque psoriasis episodes. Treatment with both systemic and topical KYC was efficacious for reducing skin lesion severity. Plaque psoriasis has variable manifestations, and not all available treatments are efficacious for patients and can often be invasive and expensive [62,63,64,65]. Currently, there are very few approved therapies that target oxidative stress in plaque psoriasis. Thus, inhibiting MPO would be an innovative therapeutic strategy that could be used in tandem with other drugs.

## Figures and Tables

**Figure 1 antioxidants-10-01338-f001:**
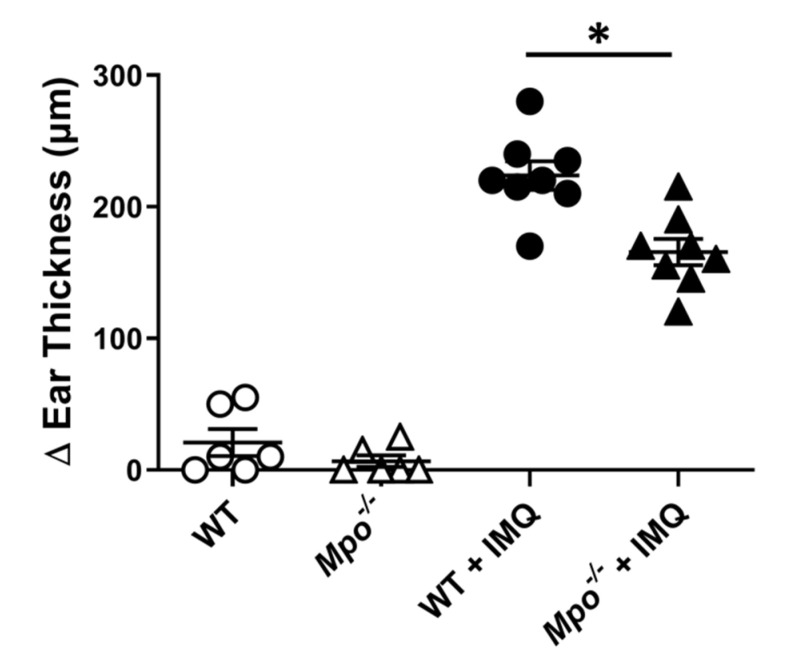
MPO deficiency ameliorates plaque psoriasis ear thickening. Plaque psoriasis was induced on the ears of wildtype C57BL/6 and *Mpo*^−/−^ mice. Ears were measured at baseline prior to induction and then again on day five. Baseline ear thickness was subtracted from the final thickness per ear, shown as change in (Δ) thickness. White circle = non-induced WT ear; white triangle = non-induced *Mpo^−/−^* ear; black circle = induced WT ear; black triangle = induced *Mpo^−/−^* ear. Data include two replicate experiments each containing 3–4 mice per group. Each data point represents one mouse. * *p* ≤ 0.05.

**Figure 2 antioxidants-10-01338-f002:**
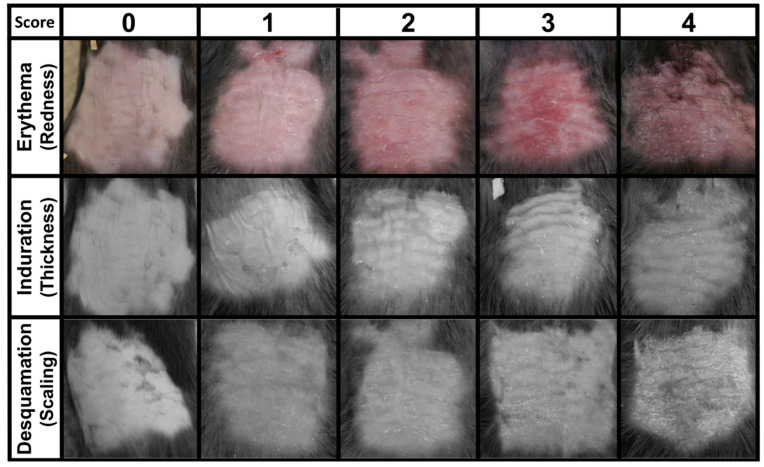
Mouse PASI scoring system. To determine overall skin inflammation during plaque psoriasis, erythema, induration, and desquamation were assessed individually using scores 0 (no incidence) to 4 (most severe) (see Table 1). Each image is unique and compiled from three replicate experiments each containing three mice.

**Figure 3 antioxidants-10-01338-f003:**
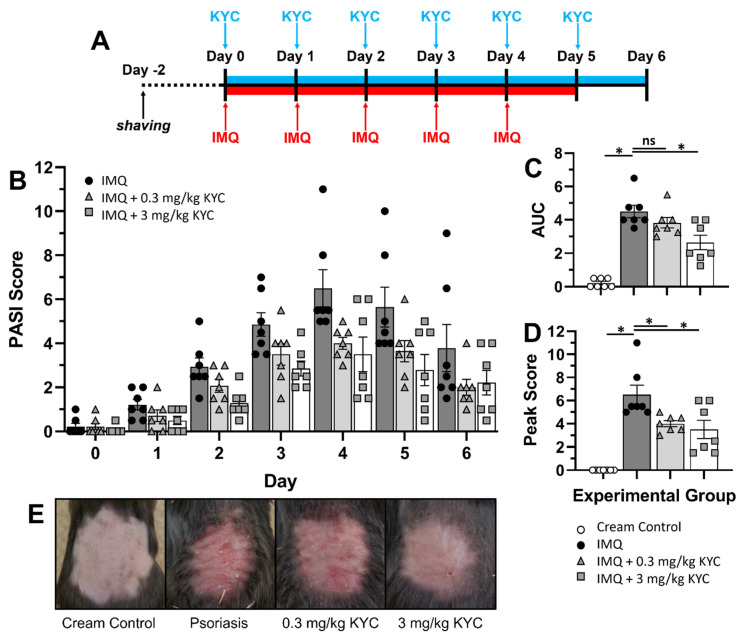
Systemic administration of KYC ameliorates plaque psoriasis in mice. Mice were shaved on the back two days before psoriasis induction. At the experiment start, mice were systemically IP administered KYC or PBS alone daily for six days and at the same time, IMQ cream was applied daily for five days. (**A**) Experimental timeline and treatment regimen is shown. (**B**) Average daily PASI scores for each condition are shown. (**C**) AUC from the entire disease course was calculated from the individual daily scores. (**D**) Peak psoriasis incidence on day 4 is shown. (**E**) Representative skin images were recorded at peak psoriasis (day 4). Cream control = white circles; IMQ psoriasis = black circles; IMQ + 0.3 mg/kg KYC = gray triangles; IMQ + 3 mg/kg KYC = gray squares. Each data point represents one mouse. Data shown are the mean and SEM (**B**–**D**). Data include two replicate experiments containing 3–4 mice per group (**A**–**D**). * *p* ≤ 0.05.

**Figure 4 antioxidants-10-01338-f004:**
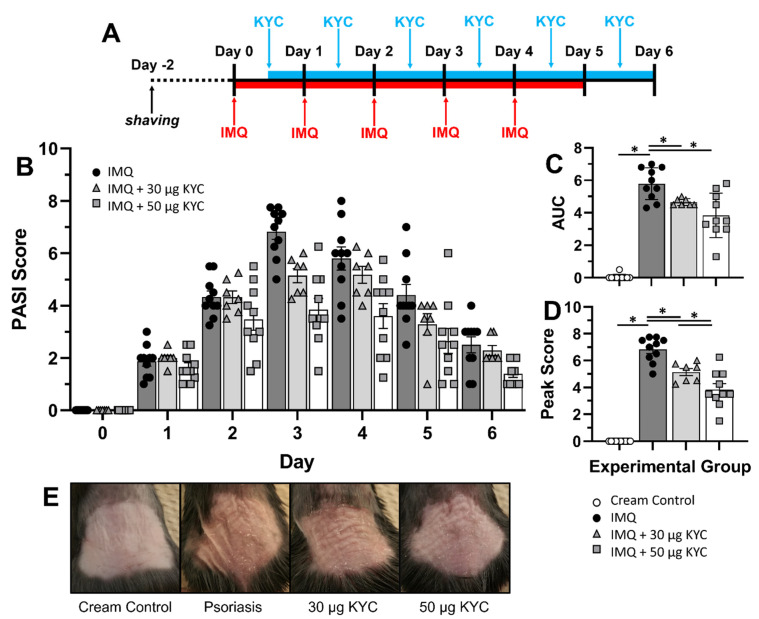
Topical administration of KYC in cream ameliorates plaque psoriasis in mice. Mice were shaved on the back two days before psoriasis induction. At the experiment start, IMQ cream was applied daily for five days in the morning (AM). KYC in cream was administered at least eight hours after IMQ once signs of psoriasis began. Mouse PASI scores were assessed prior to IMQ cream application (AM) and prior to KYC cream treatment (PM); shown are the average scores. (**A**) Experimental timeline and treatment regimen. (**B**) Composite daily PASI scores averaged between AM and PM treatments are shown. (**C**) AUC from the entire disease course was calculated from the daily scores. (**D**) Peak psoriasis incidence on Day 3 (AM + PM) was isolated for comparing treatment groups. (**E**) Representative skin images were recorded at peak psoriasis (Day 3). Cream control = white circles; IMQ psoriasis = black circles; IMQ + 0.3 mg/kg KYC = gray triangles; IMQ + 3 mg/kg KYC = gray squares. Each data point represents one mouse. Data shown are the mean and SEM (**B**–**D**). Data include two replicate experiments containing 3–4 mice per group. * *p* ≤ 0.05.

**Figure 5 antioxidants-10-01338-f005:**
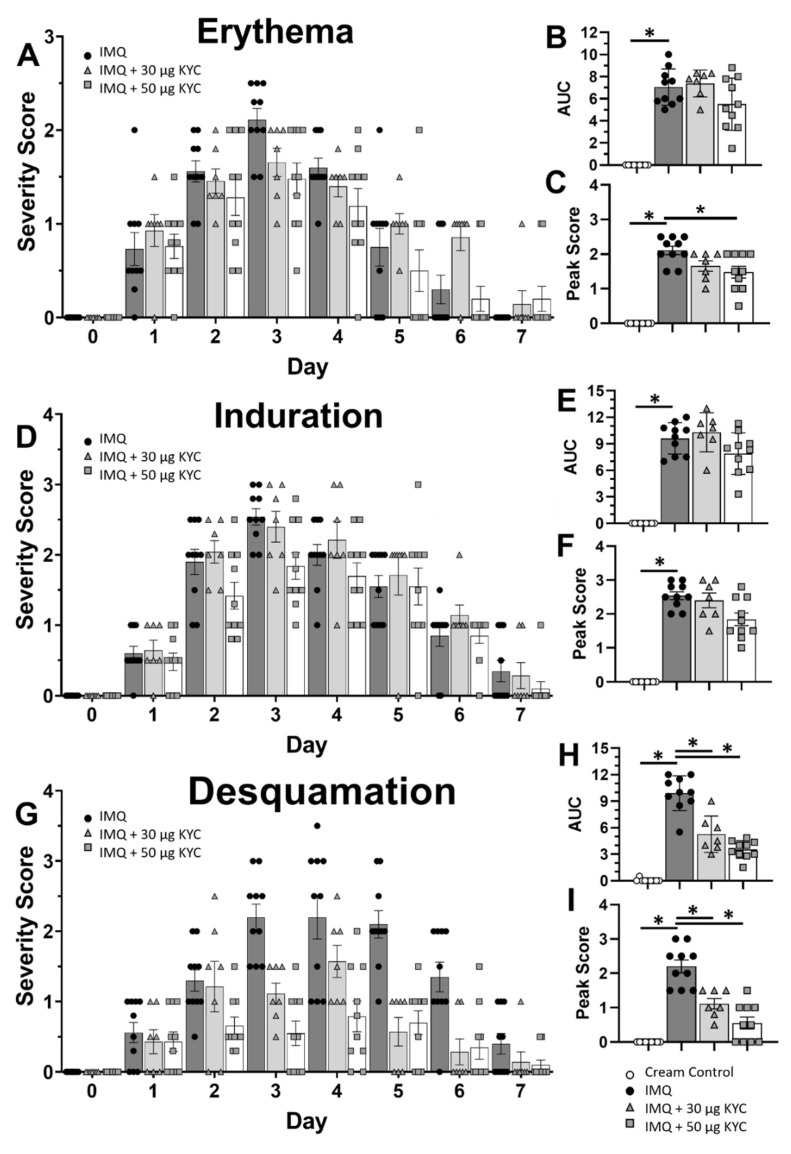
Topical application of KYC cream differentially attenuates parameters of skin inflammation. Cumulative PASI scores from Figure 4B were parsed into individual skin parameters: erythema (**A**–**C**), induration (**D**–**F**), and desquamation (**G**–**I**). (**A**,**D**,**G**) Daily PASI parameter scores are shown. (**B**,**E**,**H**) AUC from the daily scores was calculated per parameter. (**C**,**F**,**I**) Peak skin parameter incidence (day 3) is shown. Cream control = white circles; IMQ psoriasis = black circles; IMQ + 30 mg/kg KYC = gray triangles; IMQ + 50 mg/kg KYC = gray squares. Each data point represents one mouse. Data shown are the mean and SEM (**A**–**I**). Data include two replicate experiments containing 4–5 mice per group (**A**–**I**). * *p* ≤ 0.05.

**Figure 6 antioxidants-10-01338-f006:**
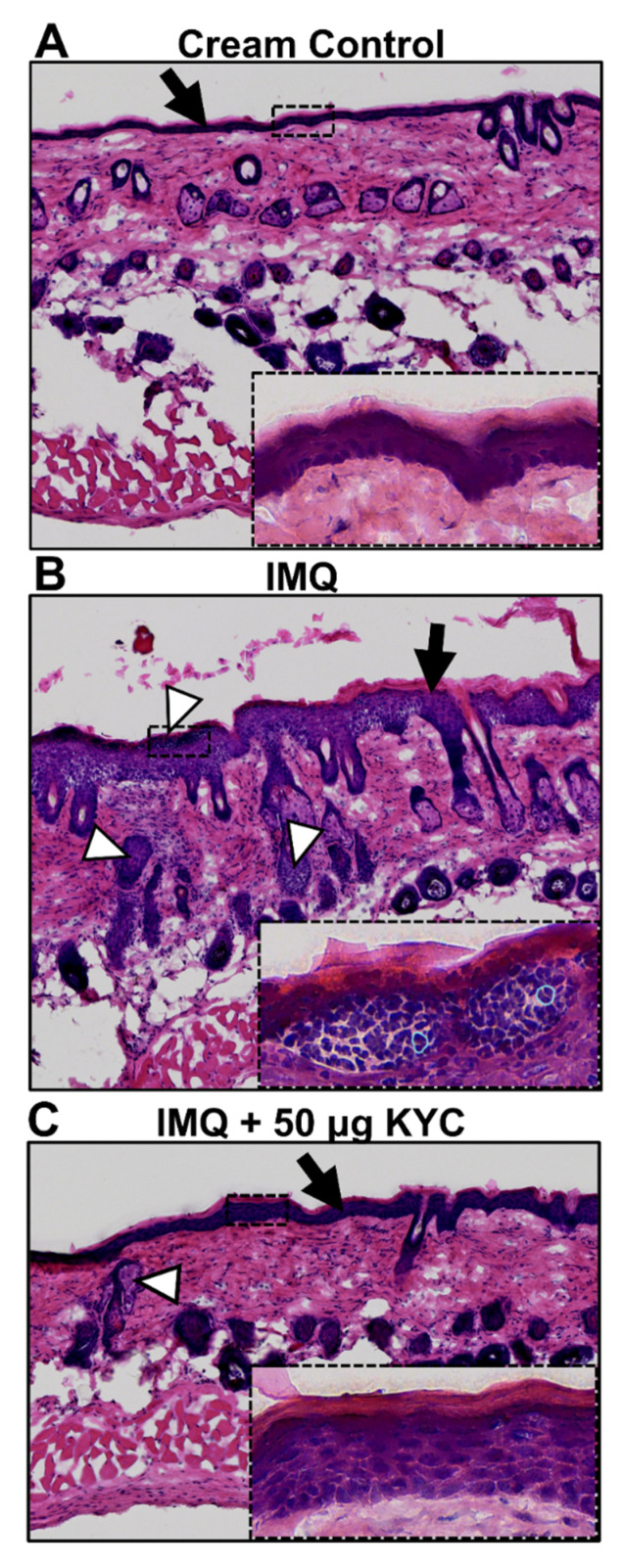
Skin histology during IMQ-induced skin inflammation and treated with KYC. Mice were administered IMQ or control cream daily for three days. Starting on day 1, KYC cream was given daily for two days. Skin sections were collected on day 3 at peak of disease, stained with H&E, and then assessed for overall signs of inflammation. Skin from a healthy, non-diseased mouse (**A**); psoriasis control mouse and (**B**) from a 50 μg KYC cream-treated mouse (**C**) are shown. Black arrows indicate the epidermal layer; white arrowheads demarcate sites of immune infiltration into the dermis; blue outlines highlight example neutrophil borders. Images shown are representative of a single mouse per treatment group.

**Table 1 antioxidants-10-01338-t001:** Description of mouse PASI scores.

Score	0	1	2	3	4
**Erythema** **(redness)**	Fleshy pink. (Minor cuts appear dark but discreet)	Minor reddening across surface.	Medium red across surface.	Medium red across surface with dark red patches.	Dark red
**Induration (thickness)**	No skin thickening. Flesh should be loose and unwrinkled	Visible skin puckering; thickness unchanged when pinched (~1 mm)	Skin thickness of 1–2 mm when pinched; visible ridges in some areas.	Skin thickness of >2mm when pinched; visible ridges are loose.	Skin thickness of >2 mm without pinching; visible ridges are tight.
**Desquamation (scaling)**	No skin flaking.	Minor dry spots without flaking.	Dry spots across a majority of the skin; flaking along crevices	Moderate flaking across a large surface area.	Moderate flaking across a large surface area; severe flaking along crevices

## Data Availability

The data presented in this study are available in article and supplementary material.

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
