# Peer review of "Myeloperoxidase Inhibition Ameliorates Plaque Psoriasis in Mice"

_antioxidants, 2021, doi:10.3390/antiox10091338_

Round 1
Reviewer 1 Report
This study by Neu et al. analyzed the effect of MPO inhibitor KYC and of MPO deficiency on IMQ-induced plaque psoriasis. They found that systemic and topical administration of KYC to the shaved back skin of IMQ-induced wild-type mice ameliorated the peak lesion severity. They also found that MPO deficiency ameliorated plaque psoriasis ear thickening. From these results, they have discussed targeting MPO oxidative stress pathways for treatment of plaque psoriasis.
Comments:
- 4, 5, and 6 showed that topical application of KYC attenuated the IMQ-induced plaque psoriasis, and the authors have concluded that the attenuation was due to inhibition of MPO by KYC. In the majority of cases, main source of MPO is believed to be neutrophils. Did the authors observe infiltration of neutrophils or accumulation of MPO to the site of KYC treatment? Identification of the type of infiltrated immune cells observed in Fig. 6 could be important.
- Same model system should be used to properly compare the effect of MPO deficiency and MPO inhibition on the plaque psoriasis. However, the ear thickening model shown in Fig.1 and the back skin model used in Figs. 2 to 6 are not necessarily the same one. It might be better to examine the IMQ-induced plaque psoriasis on the back skin of MPO-deficient mice and replace the results with those shown in Fig. 1.
Minor points:
- Lines 62-63: MPO is mainly exist in neutrophils and to a lesser degree in monocytes as shown by lots of original papers or reviews. Although MPO has also been found to be associated by some tissue-resident macrophages, such as though found in atherosclerotic lesions (refs. 28, 29), it is not certain whether this is related to expression or the uptake of neutrophil-derived MPO by the macrophage from the extracellular space.
- Line 121: please give the full name for TFA.
- Line 178: Is Ref. 44 cited correctly?
- 1, lines 179-182: Please define the white circle WT and white triangle MPO-/- mice more clearly. It is unclear whether they are steady state ear thickness or non-induced left ear control.
- Lines 235-236: days 0-5 and 0-6 could be days 0-4 and 0-5, respectively. Also, day 3 (line 252) and day 7 (line 261) could to be day 4 and day 6, respectively.
- Lines 288-289: Fig. 5C shows the significant difference between IMQ+30ug KYC group and controls, which is inconsistent with the results described in lines 288-289.
- 2, 3E, and 4E: The quality of pictures is not high enough to make any interpretation.
Author Response
Rebuttal to reviews on “Myeloperoxidase Inhibition Ameliorates Plaque Psoriasis in Mice”
We sincerely thank both reviewers for their time and consideration in assessing our manuscript. The comments and suggestions provided were valuable in editing this work, and we have considered them all and adjusted the paper accordingly. We are providing the following feedback on these revisions and hope that the reviewers and journal editors will consider the manuscript ready for publication. Changes to the manuscript are highlighted in yellow.
Reviewer 1
Comments:
- “Did the authors observe infiltration of neutrophils or accumulation of MPO to the site of KYC treatment?”
Response: High resolution images of H&E staining were taken to identify the cell types found in the IMQ skin inflammation. This image was added to Fig. 6, which clearly show neutrophil infiltration into the skin. Because KYC only inhibits MPO, we do not believe that this treatment would alter MPO presence in the skin, so we did not pursue this experiment.
- “It might be better to examine the IMQ-induced plaque psoriasis on the back skin of MPO-deficient mice and replace results with those shown in Fig. 1.”
Response: Although repeating the experimental setup from Fig. 1 with MPO-/- mice in psoriasis would be ideal, these animals are scarcely available from the vendor and we had to sacrifice our colony due to COVID. Nevertheless, a major caveat to using MPO-/- mice is that in various disease models of inflammation differential penetrance was shown. That is why we primarily use MPO inhibition for our studies. We believe this is due to differences in the gut microbiome, which is the focus of a different study.
Minor points:
- “Although MPO has also been found in atherosclerotic lesions, it is not certain whether this is related to expression or the uptake of neutrophil-derived MPO by the macrophage from the extracellular space.”
Response: Evidence from others suggests that MPO can, indeed, be generated by macrophages themselves based on expression of MPO transcripts. We have examined macrophage MPO production in other studies, which showed that macrophage MPO expression is subset specific. Unfortunately, we cannot discuss those findings here because the manuscript is in preparation.
- “TFA”
Response: The full name for this chemical has been provided (trifluoroacetic anhydride).
- “Ref 44”
Response: The title of the paper had been cited incorrectly, but this has been rectified. All other information in this citation was accurate.
- “It is unclear whether they are steady state ear thickness or non-induced left ear control.”
Response: The description of the experimental conditions were made clearer in the paper text and figure legends.
- Adjusting experimental time points
Response: We do not understand the reason for changing the timeline of the experiment. For each figure, we have provided a diagram of the timeline to make it clear to the reader. Thus, we have not altered the day numbering of the timeline.
- “Fig. 5C shows the significant difference between IMQ+30ug KYC group and controls, which is inconsistent with the results described.”
Response: Thank you for catching this error. The error bars were incorrect in the original figure and have been changed to reflect the results and statistics described.
- “The quality of pictures is not high enough to make any interpretation.”
Response: These images were taken from live animals during the experiment, so we were limited to restrictions in the animal facility for manipulating mice. We could not take the mice out of the facility to photograph them and then return them. Unfortunately, we do not have dissecting microscopes equipped with a high quality camara to take images in the animal facility. Thus, the images were taken within a flow hood/cabinet with restricted camera exposure and lighting. Animal breathing also made it difficult to focus the lens across the curvature of the back. We provided the images to show the readers that a qualitative analysis was performed to generate our PASI scoring system. To complement the photographic images, we also generated drawn images of each stage of psoriasis. These are now included in the Supplemental materials to make interpreting the scoring scale more clear.
Reviewer 2 Report
Two mouse models for psoriasis skin lesions were applied to study the contribution of myeloperoxidase in this inflammatory condition. Psoriasis-like conditions were induced by treatment with Aldara IMQ cream. IMQ-induced ear inflammation was less significant in MPO-deficient mice in contrast to wildtype ones. Skin inflammation on the back of wildtype mice was attenuated by both systemic and topical application of the myeloperoxidase inhibitor KYC. This inhibitor reduced also epithelial hypertrophy and immune cell infiltration in IMQ-treated mice.
The study is well performed. All conclusions are sound.
I have the following minor remark.
line 56: myeloperoxidase is not a protease
Author Response
Rebuttal to reviews on “Myeloperoxidase Inhibition Ameliorates Plaque Psoriasis in Mice”
We sincerely thank both reviewers for their time and consideration in assessing our manuscript. The comments and suggestions provided were valuable in editing this work, and we have considered them all and adjusted the paper accordingly. We are providing the following feedback on these revisions and hope that the reviewers and journal editors will consider the manuscript ready for publication. Changes to the manuscript are highlighted in yellow.
Reviewer 2
- “Myeloperoxidase is not a protease.”
Response: That was an embarrassing error, which has been corrected. Thank you for catching that.
Round 2
Reviewer 1 Report
I do not have any additional questions or remarks.